# Caloric and Macronutrient Intake and Meal Timing Responses to Repeated Sleep Restriction Exposures Separated by Varying Intervening Recovery Nights in Healthy Adults

**DOI:** 10.3390/nu12092694

**Published:** 2020-09-03

**Authors:** Andrea M. Spaeth, Namni Goel, David F. Dinges

**Affiliations:** 1Department of Kinesiology and Health, Division of Life Sciences, School of Arts and Sciences, Rutgers University, New Brunswick, NJ 08901, USA; 2Biological Rhythms Research Laboratory, Department of Psychiatry and Behavioral Sciences, Rush University Medical Center, Chicago, IL 60612, USA; namni_goel@rush.edu; 3Unit for Experimental Psychiatry, Division of Sleep and Chronobiology, Department of Psychiatry, University of Pennsylvania Perelman School of Medicine, Philadelphia, PA 19104, USA; dinges@pennmedicine.upenn.edu

**Keywords:** repeated sleep deprivation, energy intake, recovery sleep, timed eating, late-night eating, macronutrients, sugar, fiber, saturated fat

## Abstract

Sleep restriction (SR) reliably increases caloric intake. It remains unknown whether such intake cumulatively increases with repeated SR exposures and is impacted by the number of intervening recovery sleep opportunities. Healthy adults (33.9 ± 8.9y; 17 women, Body Mass Index: 24.8 ± 3.6) participated in a laboratory protocol. N = 35 participants experienced two baseline nights (10 h time-in-bed (TIB)/night; 22:00–08:00) followed by 10 SR nights (4 h TIB/night; 04:00–08:00), which were divided into two exposures of five nights each and separated by one (n = 13), three (n = 12), or five (n = 10) recovery nights (12 h TIB/night; 22:00–10:00). Control participants (n = 10) were permitted 10 h TIB (22:00–08:00) on all nights. Food and drink consumption were ad libitum and recorded daily. Compared to baseline, sleep-restricted participants increased daily caloric (+527 kcal) and saturated fat (+7 g) intake and decreased protein (−1.2% kcal) intake during both SR exposures; however, intake did not differ between exposures or recovery conditions. Similarly, although sleep-restricted participants exhibited substantial late-night caloric intake (671 kcal), such intake did not differ between exposures or recovery conditions. By contrast, control participants showed no changes in caloric intake across days. We found consistent caloric and macronutrient intake increases during two SR exposures despite varying intervening recovery nights. Thus, energy intake outcomes do not cumulatively increase with repeated restriction and are unaffected by recovery opportunities.

## 1. Introduction

Despite recommendations that adults habitually obtain 7–9 h of sleep per night [1], approximately 35% of American adults report typically sleeping less than 7 h per night [2]. This type of chronic partial sleep restriction leads to an accumulation of sleep debt over time accompanied by progressive increases in daytime sleepiness and decrements in cognitive and physiological functioning [3,4,5]. Furthermore, chronic sleep restriction has been associated with increased risk for cardiovascular disease, obesity, diabetes, and overall morbidity and mortality [1,5,6,7], and the Center for Disease Control recently recognized chronic sleep restriction as a considerable public health issue [8]. A common pattern practiced by most adults is to curtail sleep during the work/school week, and then attempt to “catch-up” on sleep during weekends [9,10]. As such, there is growing interest in understanding how repeated exposures to sleep restriction, separated by varying nights of extended recovery sleep, impact waking function [11,12,13,14,15].

A recent study measured psychomotor vigilance test (PVT) performance across three cycles of sleep restriction (two consecutive nights of 3 h time-in-bed (TIB)) and recovery sleep (one night, 10 h TIB) [16]. During the first cycle, PVT performance was significantly impaired during sleep restriction and nearly returned to baseline levels after recovery sleep. When re-challenged with subsequent exposure to sleep restriction, PVT performance demonstrated a cumulative impairment that worsened at increasingly faster rates, suggesting that the apparent nearly-complete recovery was temporary and deficits from re-exposure were cumulative [16]. Similarly, Mullington and colleagues investigated blood pressure and stress markers (cortisol, the inflammatory marker IL-6, glucocorticoid receptor sensitivity) across four cycles of sleep restriction (three consecutive nights of 4 h TIB) and recovery sleep (one night 8 h TIB) [17,18]. Compared to baseline, sleep restriction adversely affected blood pressure and stress markers, with repeated exposure having a cumulative deleterious effect on these markers, particularly cortisol [17,18].

In adults, habitual insufficient sleep (<7 h/night) is a significant risk factor for weight gain and obesity [1,6,19], and experimental studies have demonstrated that a single exposure of sleep restriction (five nights, 4–5 h/night) leads to significant increases in hunger, late-night eating, daily caloric intake, fat intake, and weight gain [20,21,22,23,24]. However, to our knowledge, only one study, by Depner and colleagues, has assessed the impact of repeated exposures of sleep restriction with intervening recovery sleep on caloric intake in humans [25]. In this study, participants experienced sleep restriction (four consecutive nights, 5 h TIB/night) followed by two nights of ad libitum weekend recovery sleep (10 h TIB minimum/night, napping permitted) followed by a shorter second exposure of sleep restriction (two consecutive nights, 5 h TIB/night) [25]). Daily intake and after-dinner intake were increased during both exposures of sleep restriction to a comparable degree and did not differ from the intake of participants who experienced eight consecutive nights of sleep restriction (5 h TIB/night), suggesting that the intervening two recovery sleep nights did not protect against the effects of subsequent sleep restriction on intake. Studies examining other types of waking function have also suggested that two nights of recovery sleep is insufficient for fully restoring sleep debt and waking behavior [26,27,28,29].

The purpose of the current study was to examine the effects of two identical-length exposures of sleep restriction on caloric and macronutrient intake and late-night eating while systematically varying and increasing the number of intervening recovery nights. We aimed to determine if intake during a second exposure of sleep restriction differed between participants receiving one, three, or five nights of 12 h recovery sleep opportunities between sleep restriction exposures. We hypothesized that participants in the control group (10 h TIB/night) would not exhibit changes in intake across protocol days whereas sleep-restricted participants would increase caloric intake and fat intake, and exhibit late-night intake during both exposures of sleep restriction (each exposure: five consecutive nights, 4 h TIB/night). We further hypothesized that intake during the second exposure would be cumulatively greater than during the first exposure and that participants who received fewer nights of intervening recovery sleep would exhibit greater cumulative intake changes during the second sleep restriction exposure than those who received more nights of intervening recovery sleep. Given our previous studies demonstrating changes in fiber, sugar, and saturated fat intake during sleep restriction [30,31], we also explored variations in these nutrients during the protocol.

## 2. Materials and Methods

The protocol was approved by the University of Pennsylvania’s Institutional Review Board (IRB number: 810896), and all participants were compensated for their participation. All participants provided written informed consent in accordance with the Declaration of Helsinki.

### 2.1. Participants

Healthy adults, aged 22–50 years, were recruited in response to study advertisements. They were typical sleepers, reporting 6.5 h to 8.5 h of sleep per night with bedtimes between 22:00 and 00:00, and wake times between 06:00 and 09:00. They had no evidence of habitual napping or sleep disturbances (i.e., no complaints of insomnia, daytime sleepiness, or other sleep–wake disturbances), and they did not exhibit extreme morningness or extreme eveningness, as assessed by the Composite Scale of Morningness and Eveningness [32]. They were free of acute and chronic medical and psychological conditions, as established by interviews, clinical history, questionnaires, physical examinations, and blood (including a fasting blood glucose test) and urine tests. Participants were nonsmokers, normal to overweight, and did not participate in shift work, transmeridian travel, or irregular sleep–wake routines in the 60 days prior to the study. Sleep disorders were further screened by a night of in-laboratory polysomnography and oximetry measurements.

### 2.2. Protocol

Participants were not permitted to use caffeine, alcohol, tobacco, or medications (except oral contraceptives) in the week before the laboratory experiment, as verified by blood and urine screenings. During the weeks before and after the in-laboratory, participants were monitored at home with actigraphy, sleep–wake diaries, and time-stamped call-ins to assess sleep duration, bedtime, and wake time. For the in-laboratory phase of the protocol, participants were admitted to the Sleep and Chronobiology Laboratory at the Hospital of the University of Pennsylvania (HUP). Participants were studied for 18–20 consecutive days continuously with daily clinical checks of vital signs and symptoms by nurses and an independent physician on call. Participants were randomized as a group (n = 4 per group) to one of the four following conditions: 1. control (CON); 2. one night of recovery sleep between two exposures of sleep restriction (1 REC); 3. three consecutive nights of recovery sleep between two exposures of sleep restriction (3 REC); or 4. five consecutive nights of recovery sleep between two exposures of sleep restriction (5 REC). In all three experimental conditions, each sleep restriction exposure involved five consecutive nights of sleep restricted to 4 h time-in-bed (TIB) per night (from 04:00 to 08:00) and each night of recovery sleep involved 12 h TIB per night (from 22:00 to 10:00). Figure 1 depicts an overview of the full study protocol. We selected a sleep restriction paradigm consisting of 4 h TIB for five consecutive nights since this degree of sleep loss produces cumulative neurobehavioral deficits in most healthy adults [3,4,33,34,35] and is within the range of sleep loss that occurs as a result of societal or lifestyle factors [2,3,9]. The control condition involved similar procedures as the experimental conditions except that participants were allowed 10 h TIB each night for sleep (from 22:00 to 08:00) during the in-laboratory stay.

During the in-laboratory phase of the study, participants were not permitted to leave the laboratory. Participants were ambulatory but were not allowed to exercise. Participants were permitted to watch television, read, play video or board games, and perform other sedentary activities between protocol procedures. The light levels in the laboratory were held constant at <50 lux during scheduled wakefulness and <1 lux during scheduled sleep periods. Ambient temperature was maintained between 22 and 24 °C. Participants were behaviorally monitored by trained staff continuously throughout the protocol to ensure adherence.

### 2.3. Measures and Procedures

Upon admittance to the in-laboratory protocol, nurses measured each participant’s height and weight (while participants wore minimal clothing and no shoes) at the Center for Human Phenomic Science (formally the Clinical and Translational Research Center) at HUP using a calibrated scale. Participants provided their age, race, and gender via self-report questionnaires.

Participants selected their meals/snacks by choosing from various menu options, selecting additional food/drink available in the kitchen within the laboratory suite (which included a refrigerator, microwave, and toaster oven) and by making requests to the monitors or study coordinator. In order to ensure that participants were provided sufficient time to eat each day, three 30- to 45-min opportunities were specified in the protocol during days with a 22:00 bedtime (at 09:00, 12:35, and 18:30) and one additional 30-min opportunity to eat was specified in the protocol during days with a 04:00 bedtime (at 00:30). Participants could choose to eat or not eat during these specified periods. In addition to these specified mealtimes, participants could consume food/drink at any time while awake during the protocol other than when they were completing neurobehavioral tests or were being fasted for an energy expenditure measurement [36]. Participants were never told that they had to eat/drink, and they were instructed to eat/drink whenever they wanted if it did not interfere with testing times. Participants were also instructed that they could eat what they had ordered or could select from other foods available in the laboratory kitchen and that they should eat as much (or as little) as they preferred. Participants retrieved their own food/drink from the kitchen inside the laboratory suite whenever they wanted to eat/drink and could eat at a table in the common area or privately in their bedrooms. Participants were not permitted to consume caffeinated beverages or chocolate during the protocol.

All food was weighed and recorded prior to being provided to the participants. To enhance the measurement accuracy of each food’s weight, food was provided in individual containers. Each day, a detailed description of the items, the amount consumed, and the time each item was consumed were recorded by trained monitors. Additionally, any food/drink that was left over after each meal was weighed and recorded. Intake data were entered into The Food Processor SQL program (version 10.11; ESHA Research, Salem, OR), which is a validated professional nutrition analysis software and database program [37] that provides components of food and drink intake including calories and macronutrients, and that we have used in several prior studies [21,22,30,31,36,38].

Consistent with our previous studies [21,22], caloric and macronutrient intake from wake time (08:00) to 21:59 during the two days following baseline sleep were averaged and defined as baseline caloric intake. Caloric and macronutrient intake from wake time (08:00) to 03:59 during sleep restriction days 1–3 were averaged and defined as “SR Exposure 1” intake. Caloric and macronutrient intake from wake time (08:00) to 03:59 during sleep restriction days 6–8 were averaged and defined as “SR Exposure 2” intake. Although caloric and macronutrient intake were measured during SR4 and SR5, these data were not included in the current analyses because participants were fasted during SR4 as part of a separate experiment [34]. To be consistent with SR Exposure 1, only three days of such data were included in the average for SR Exposure 2 (SR9 and SR10 were excluded). Data for control condition days (CD) 7–13 were averaged for comparison with corresponding experimental condition days SR6-8 since this varied between experimental conditions (Figure 1).

### 2.4. Data Analysis

Mixed-model ANOVAs compared caloric intake, macronutrient intake, and meal timing changes across protocol days first between the control and experimental (all sleep-restricted participants) conditions, and then across the three experimental conditions (1 REC, 3 REC, and 5 REC). If significant Day or Day*Condition effects were observed, planned comparisons were conducted using paired-samples t-tests. Effect sizes were calculated using Cohen′s d (small, d = 0.2; medium, d = 0.5; large, d = 0.8) [39]. Intraclass correlation coefficients (ICC: two-way mixed, absolute agreement, average measures) assessed the inter-individual differences and intra-individual stability of intake responses to the two sleep restriction exposures. The following established ranges characterize ICCs and reflect the increasing stability of observed inter-individual differences: 0.0–0.2 (slight); 0.2–0.4 (fair); 0.4–0.6 (moderate); 0.6–0.8 (substantial); and 0.8–1.0 (almost perfect) [40]. Statistical analyses were conducted using IBM SPSS Statistics for Windows (version 26).

## 3. Results

### 3.1. Participant Characteristics

Of the 55 participants who enrolled in the study, 49 participants completed the protocol. The six non-completers were either withdrawn because of protocol noncompliance (n = 1) or withdrew because of health or personal issues unrelated to the protocol (n = 5). Four participants who completed the study were not included in the analyses as they experienced mild health issues during the study that affected their appetite and intake. Therefore, the final sample included N = 45 men and women. Of these 45 participants, n = 10 were randomized to the CON condition, n = 13 were randomized to the 1 REC condition, n = 12 were randomized to the 3 REC condition, and n = 10 were randomized to the 5 REC condition. There were no significant differences between conditions in terms of demographic information, caloric need, chronotype, or pre-study actigraphic sleep (Table 1).

### 3.2. Daily Caloric Intake

A 3 (Day: BL1-2, SR1-3, SR6-8) × 2 (Condition: Control vs. Experimental) mixed-model ANOVA revealed a significant main effect of Day (F(2,86) = 6.83, *p* = 0.002) and a significant Day*Condition interaction (F(2,86) = 8.51, *p* < 0.001) for caloric intake (Figure 2A). A 3 (Day: BL1-2, SR1-3, SR6-8) × 3 (Experimental Condition: 1 REC, 3 REC, 5 REC) mixed-model ANOVA revealed a significant main effect of Day for caloric intake (F(2,64) = 28.11, *p* < 0.001); however, the Day*Experimental Condition interaction was not significant (*p* = 0.27; Figure 2B). Compared to baseline, sleep-restricted participants increased caloric intake during both exposures of sleep restriction (SR1-3: t(34) = 7.79, *p* < 0.001, d = 1.3; SR6-8: t(34) = 4.96, *p* < 0.001, d = 0.84). Caloric intake did not differ between sleep restriction exposures (*p* = 0.18) and the ICC comparing caloric intake between the first and second exposures of sleep restriction was 0.95 (almost perfect). Participants in the control condition did not exhibit changes in caloric intake across corresponding protocol days (*p* = 0.34; Figure 2A).

### 3.3. Daily Macronutrient Intake

When examining daily macronutrient intake (Table 2), a 3 (Day: BL1-2, SR1-3, SR6-8) × 2 (Condition: Control v. Experimental) mixed-model ANOVA revealed a significant main effect of Day for the percentage of calories consumed from protein (F(2,86) = 4.81, *p* = 0.010); however, the Day*Condition interaction was not significant (*p* = 0.83). A 3 (Day: BL1-2, SR1-3, SR6-8) × 3 (Experimental Condition: 1 REC, 3 REC, 5 REC) mixed-model ANOVA revealed a significant main effect of Day for the percentage of calories consumed from protein (F(2,64) = 6.76, *p* = 0.002); however, the Day*Experimental Condition interaction was not significant (*p* = 0.15). Compared to baseline, sleep-restricted participants decreased the percentage of calories consumed from protein during both exposures of sleep restriction (SR1-3: t(34) = 3.19, *p* = 0.003, d = 0.54; SR6-8: t(34) = 2.41, *p* = 0.02, d = 0.41). Protein intake did not differ between sleep restriction exposures (*p* = 0.10) and the ICC comparing protein intake between the first and second exposures of sleep restriction was 0.80 (substantial). Participants in the control condition did not exhibit changes in the percentage of calories consumed from protein across corresponding days (Table 2; *p* = 0.36). Mixed-model ANOVAs revealed no significant main effects of Day or Day*Condition/Experimental Condition interactions for the percentage of calories consumed from carbohydrates (*p*’s > 0.15) or fat (*p*’s > 0.53). The ICCs comparing carbohydrate and fat intake between the first and second exposures of sleep restriction were substantial at 0.69 and 0.75, respectively.

### 3.4. Fiber, Sugar, and Saturated Fat Intake

When examining fiber intake (Table 3), 3 (Day: BL1-2, SR1-3, SR6-8) × 2 (Condition: Control v. Experimental) mixed-model ANOVAs revealed a significant main effect of Day (F(2,86) = 5.37, *p* = 0.006) for grams (g) consumed and a Day*Condition interaction effect (F(2,86) = 4.06, *p* = 0.021) for g/1000 kcal. The 3 (Day: BL1-2, SR1-3, SR6-8) × 3 (Experimental Condition: 1 REC, 3 REC, 5 REC) mixed-model ANOVAs revealed a significant main effect of Day (F(2,64) = 3.45, *p* = 0.038) for grams consumed and a significant Day*Experimental Condition interaction effect (F(4,64) = 2.96, *p* = 0.026) for g/1000 kcal. Compared to baseline, sleep-restricted participants consumed more grams of fiber during the first sleep restriction exposure (t(34) = 2.34, *p* = 0.03, d = 0.40), but g fiber/1000 kcal was not significantly different (*p* = 0.10). Compared to baseline, sleep-restricted participants consumed a comparable number of grams of fiber during the second sleep restriction exposure (*p* = 0.16), and although g fiber/1000 kcal was lower, the difference did not reach significance (*p* = 0.07). Intake of fiber (grams and g/1000 kcal) did not differ between sleep restriction exposures (*p*’s > 0.10) and the ICCs comparing fiber intake between the first and second exposures of sleep restriction were 0.87 (grams, almost perfect) and 0.86 (g/1000 kcal, almost perfect). Control participants consumed more fiber during days corresponding with SR1-3 than during baseline (grams: t(9) = 2.94, *p* = 0.02; g/1000 kcal: t(9) = 4.06, *p* = 0.026) and than during days corresponding with SR6-8 (grams: t(9) = 3.50, *p* = 0.007; g/1000 kcals: t(9) = 2.09, *p* = 0.066). Fiber intake did not differ between baseline and days corresponding to SR6-8 (*p*’s > 0.30).

When examining sugar intake (Table 4), 3 (Day: BL1-2, SR1-3, SR6-8) × 2 (Condition: Control v. Experimental) mixed-model ANOVAs revealed a significant Day*Condition interaction (F(2,86) = 5.56, *p* = 0.005) for grams consumed and a significant main effect of day (F(2,86) = 6.28, *p* = 0.003) for percentage of daily calories. The 3 (Day: BL1-2, SR1-3, SR6-8) × 3 (Experimental Condition: 1 REC, 3 REC, 5 REC) mixed-model ANOVAs revealed significant main effects of Day for grams consumed (F(2,64) = 8.47, *p* = 0.001) and percentage of daily calories (F(2,64) = 5.34, *p* = 0.007). Compared to baseline, sleep-restricted participants consumed more grams of sugar during the first sleep restriction exposure (t(34) = 4.23, *p* < 0.001, d = 0.72), but the percentage of calories consumed from sugar was not significantly different (*p* = 0.43). Compared to baseline, sleep-restricted participants consumed a comparable amount of grams of sugar during the second sleep restriction exposure (*p* = 0.14), but the percentage of calories consumed from sugar was significantly lower (t(34) = 2.77, *p* = 0.009, d = 0.47). Intake of sugar was higher during the first sleep restriction exposure than the second sleep restriction exposure (grams: t(34) = 3.24, *p* = 0.003, d = 0.56; % kcal: t(34) = 2.97, *p* = 0.005, d = 0.50). The ICCs comparing sugar intake between the first and second exposures of sleep restriction were 0.94 (grams, almost perfect) and 0.87 (% kcal, almost perfect). In control participants, sugar intake was significantly lower during days corresponding to SR6-8 (grams: t(9) = 2.86, *p* = 0.02; % kcal: t(9) = 3.02, *p* = 0.014) than baseline. Sugar intake did not differ across other days (*p*’s > 0.06).

When examining grams of saturated fat intake (Table 5), a 3 (Day: BL1-2, SR1-3, SR6-8) × 2 (Condition: Control v. Experimental) mixed-model ANOVA revealed a significant main effect of Day (F(2,86) = 3.59, *p* = 0.032) and a Day*Condition interaction that approached significance (F(2,86) = 3.07, *p* = 0.051). The 3 (Day: BL1-2, SR1-3, SR6-8) × 3 (Experimental Condition: 1 REC, 3 REC, 5 REC) mixed-model ANOVA revealed a significant main effect of Day (F(2,64) = 9.33, *p* < 0.001); however, the Day*Experimental Condition interaction was not significant (*p* = 0.10). Mixed-model ANOVAs revealed no significant main effects or interaction effects for saturated fat as a percentage of daily calories. Sleep-restricted participants consumed more grams of saturated fat during both sleep restriction exposures (SR1-3: t(34) = 4.80, *p* < 0.001, d = 0.81; SR6-8: t(34) = 3.04, *p* = 0.004, d = 0.52) than baseline. Intake of saturated fat did not differ between sleep restriction exposures (*p* = 0.25). The ICCs comparing saturated fat intake between the first and second exposures of sleep restriction were 0.86 (grams, almost perfect) and 0.81 (% kcal, substantial). Saturated fat intake did not vary across corresponding days among participants in the control condition (*p*’s > 0.10).

### 3.5. Late-Night Intake

To assess late-night eating, we measured calories consumed during the additional six hours that sleep-restricted participants remained awake (22:00–03:59; Figure 3). The 2 (Day: SR1-3, SR6-8) × 3 (Experimental Condition: 1 REC, 3 REC, 5 REC) mixed-model ANOVAs revealed no significant main effects of Day or Day*Experimental Condition interactions for calories consumed from 22:00-03:59 (*p*’s > 0.40) or the percentage of daily calories consumed from 22:00–03:59 (*p*’s > 0.34). The ICCs comparing late-night intake between the first and second exposures of sleep restriction were almost perfect, 0.87 (kcal), and substantial, 0.65 (% kcal). To determine if the increase in daily caloric intake was due to late-night intake, caloric intake from 08:00–14:59 and from 15:00–21:59 was also compared across protocol days. Within-subjects ANOVAs revealed significant differences across days for 08:00–14:59 intake (F(2,68) = 14.92, *p* < 0.001) but no differences in 15:00–21:59 intake (*p* = 0.10). Compared to baseline, participants consumed fewer calories from 08:00–14:59 during the first (t(34) = 3.44, *p* = 0.002) and second (t(34) = 5.31, *p* < 0.001) exposures to sleep restriction. Therefore, the increase in daily intake during both exposures to sleep restriction (SR1-3: +575 kcal, SR6-8: +480 kcal) is largely due to late-night intake (SR1-3: 658 kcal, SR6-8: 684 kcal). Intake from 08:00–14:59 and from 15:00–21:59 did not vary across corresponding days in control participants.

Mixed-model ANOVAs revealed no significant main effects of Day or Day*Experimental Condition interactions for the percentage of calories consumed from protein, carbohydrates, or fat from 22:00–03:59 (*p*’s > 0.12). The ICCs for protein, carbohydrates, and fat intake from 22:00–03:59 between sleep restriction exposures were 0.72, 0.23, and 0.10, respectively, ranging from substantial to fair/slight. Mixed-model ANOVAs also revealed no significant main effects of Day or Day*Experimental Condition interactions for grams of fiber, sugar, or saturated fat consumed from 22:00–03:59 (*p*’s > 0.27). The ICCs for fiber, sugar, and saturated fat intake from 22:00–03:59 between sleep restriction exposures were 0.69, 0.83, and 0.84, respectively, ranging from substantial to almost perfect.

### 3.6. Intake During Recovery

Although not the central study question, for informative purposes, we used paired-samples t-tests to compare recovery days with baseline for each experimental condition using the false discovery rate (FDR) to correct for multiple comparisons (nine comparisons per measure). When applicable, within-subjects ANOVAs compared intake across recovery days. Although intake was lower during recovery days compared to baseline (Figure 2B), the only comparison that reached statistical significance after FDR correction was for the first day of recovery in the 3 REC condition (*p* < 0.05). Within-subjects ANOVAs showed no significant differences in intake across recovery days in the 3 REC or 5 REC conditions (*p*’s > 0.10). Compared to baseline, participants in the 1 REC condition consumed more calories from carbohydrates (*p* < 0.05, Table 6) during recovery and participants in the 3 REC condition consumed fewer calories from fiber on the second recovery day (*p* < 0.05; data not shown). There were no other differences in macronutrient intake or intake of fiber, sugar, or saturated fat between recovery days and baseline or across recovery days (*p*’s > 0.05). See Table 6 for macronutrient values during recovery.

## 4. Discussion

In support of our first hypothesis, we found that healthy adults increased daily caloric intake by nearly 20%, consumed 7 additional grams of saturated fat per day, and consumed over 600 calories during the late-night period (22:00–03:59) during both sleep restriction exposures. Contrary to our second and third hypotheses, the magnitude of these responses was consistent across sleep restriction exposures and between recovery conditions. Our findings are consistent with the study by Depner and colleagues that showed caloric intake and other metabolic markers were disrupted during two exposures to sleep restriction separated by two nights of ad libitum recovery sleep [25].

Studies using rodent models have also examined metabolic responses to repeated cycles of sleep restriction followed by recovery sleep [42,43,44,45]. Barf and colleagues [44] measured behavioral and metabolic outcomes during a four-week protocol consisting of five days of sleep restriction alternating with two days of recovery sleep. The rats exhibited hyperphagia during sleep restriction and the magnitude of hyperphagia increased with each subsequent sleep restriction exposure, whereas caloric intake returned to normal levels during each intervening recovery period [44]. Everson and colleagues measured behavioral and physiological outcomes during a 72-day protocol (10–13% of the animal’s lifespan) consisting of 10 days of sleep restriction alternating with two days of recovery sleep with rats fed a high-fat diet to better mimic the modern human environment [45]. This protocol also initiated marked metabolic dysfunction including hyperphagia during sleep restriction; again, the magnitude of hyperphagia increased with each subsequent sleep restriction exposure, whereas caloric intake was reduced during each recovery period [45]. Analyses of waste products and peripheral tissues revealed that the increased amount of food consumed was metabolized, likely via a lengthening of the intestine to allow for a greater surface area to absorb additional water and nutrients. Everson and others have posited that more energy is required to sustain internal functioning (i.e., digesting, absorbing, and processing food) at the tissue and cellular level when sleep is restricted and that this additional energy requirement prompts the seemingly adaptive hyperphagic response [45,46]. It is unclear why the hyperphagic response would cumulatively increase in magnitude with repeated exposure to sleep restriction in rodents but not in humans. Future studies examining caloric intake in response to more than only two sleep restriction exposures are needed in humans, though these lengthy protocols are very challenging and cost-prohibitive to implement.

The results of animal and human studies both demonstrate that the body does not adapt to repeated exposure to sleep restriction; instead, the hyperphagic response is initiated with each exposure. This is particularly concerning for humans who live in high-income countries with obesogenic environments in which calorically dense unhealthy foods are easily accessible and lifestyles are increasingly sedentary [47]. When healthy adults curtail their sleep each week, the accompanying increase in caloric intake can lead to positive energy balance and weight gain over time. Increased consumption of saturated fat and late-night (delayed) meal timing, which our participants exhibited during both sleep restriction exposures, have also been independently identified as risk factors for weight gain [48,49]. Indeed, population studies consistently demonstrate an association between chronic sleep restriction and obesity [6,19].

Although previous studies have shown that recent sleep history (e.g., banking sleep, sleep extension) affects subsequent neurobehavioral responses to sleep restriction [50,51,52], we did not observe differences in caloric and macronutrient intake responses between participants randomized to receive one, three, or five nights of recovery sleep between exposures. In a previous study [21], we noted that intake responses to sleep restriction were linked to hours of additional wakefulness rather than recent sleep history. We observed that healthy adults increased caloric intake on the first day of sleep restriction (when participants were well-rested but kept awake for 20 h (until 04:00)) but not on the fifth day following sleep restriction (when participants were sleep-deprived but only awake for 14 h and had a required bedtime of 22:00) [21]. Our prior results are consistent with the findings from the current study: intake during the second exposure to sleep restriction was similar regardless of recent sleep history. In the five nights preceding the second exposure to sleep restriction, participants randomized to 1 REC experienced one night of recovery sleep and four nights of sleep restriction, participants randomized to 3 REC experienced three nights of recovery sleep and two nights of sleep restriction, and participants randomized to 5 REC experienced five nights of recovery sleep. Furthermore, in the current study and in our previous study [21], we observed that the increased daily caloric intake during sleep restriction was accounted for by the additional calories consumed during the late-night period, when participants experienced additional hours of wakefulness. Collectively, these findings suggest that a delayed bedtime and/or number of hours of wakefulness may be better predictors of caloric intake than sleep duration the preceding night(s).

Consistent with our previous studies [30,38], there was considerable variability in the caloric intake response during sleep restriction in the current study. The change in daily caloric intake from baseline to sleep restriction ranged from −304.09 to 1638.90 kcals during the first exposure and from −514.47 to 1987.44 kcals during the second exposure. Similarly, the change in late-night caloric intake ranged from 83.32 to 1730.76 kcals during the first exposure and from 122.09 to 1301.46 kcals during the second exposure. Despite these considerable differences between participants, the response within each participant across both exposures was stable with ICCs indicating substantial to nearly perfect reliability for daily intake and late-night intake. Previously, we found that healthy adults exhibited consistent responses to five nights of sleep restriction and one night of total sleep deprivation separated by four nights of recovery sleep [30] and to two sleep restriction exposures separated by a long time interval (i.e., >2 months) [38]. Collectively, these findings highlight the importance of identifying biomarkers for vulnerability to the energy balance effects of sleep restriction and the need for countermeasures to mitigate the hyperphagia exhibited by those who are most vulnerable.

Our study had a few limitations that prompt future study. The participants were all healthy, between the ages of 21 and 50 years old, with BMIs in the normal to overweight range, and were habitually ~8 h/night sleepers. As such, our results may not be generalizable to other groups, such as adolescents, obese individuals, the elderly, or short or long sleepers. Future research should determine how individuals with obesity or other metabolic disorders respond to workweek sleep restriction with weekend recovery sleep in terms of caloric intake, macronutrient intake, and meal timing. In addition, the current study was a between-subjects design with relatively small sample sizes in each randomized condition group. Future studies are needed to confirm these findings in larger, more representative samples; the use of a within-subjects, cross-over design would improve statistical power and control for the large inter-individual differences we have observed in responses to sleep restriction [30,35,38,53]. However, as previously mentioned, these protocols using within-subjects design are difficult to implement and costly given the lengthy time commitment required. Larger sample sizes would also allow for the examination of gender and race differences in response to repeated cycles of sleep restriction and recovery sleep. We and others have found that men are at greater risk for hyperphagia and late-night eating during sleep restriction [21,22,25] and African Americans are at greater risk for weight gain during sleep restriction [21,36,54], but potential gender and race differences during repeated exposures to sleep restriction have not been systematically investigated. Previously, we [36] and others [24] have shown that sleep restriction affects energy expenditure; however, we did not measure energy expenditure throughout the protocol in the current study. Future studies are needed to examine how repeated exposure to sleep restriction impacts resting metabolic rate and physical activity. We also did not record if our participants ate alone or with other participants during the protocol; some findings suggest that social cues [55] may influence the amount of food consumed. Therefore, future studies are needed to determine if these social cues interact with sleep restriction to affect energy intake. Finally, we restricted sleep by delaying bedtime to 04:00, which may have caused circadian disruption that contributed to the observed energy intake effects. Notably, however, other studies have restricted sleep by delaying bedtime and advancing wake time (and therefore preserving sleep midpoint) and have observed comparable energy intake responses to those in our study [56]. Future work is needed to better understand how the timing of sleep and circadian markers associate with caloric intake and eating behavior.

## 5. Conclusions

Obtaining extra sleep during the weekend to recover from lost sleep during the workweek is a common self-selected sleep strategy employed by adults. We found that healthy adults exhibited increased caloric intake, had greater consumption of saturated fat, and had substantial late-night eating during repeated exposures to sleep restriction regardless of the number of intervening recovery sleep nights. Importantly, our findings indicate humans do not acclimate to sleep restriction with repeated exposures and that one, three or five nights of catch-up sleep is not sufficient for preventing future hyperphagic responses to sleep restriction. These novel findings are timely, given the worldwide obesity epidemic and the large percentage of adults who consistently sacrifice sleep time for other waking behaviors.

## Figures and Tables

**Figure 1 nutrients-12-02694-f001:**
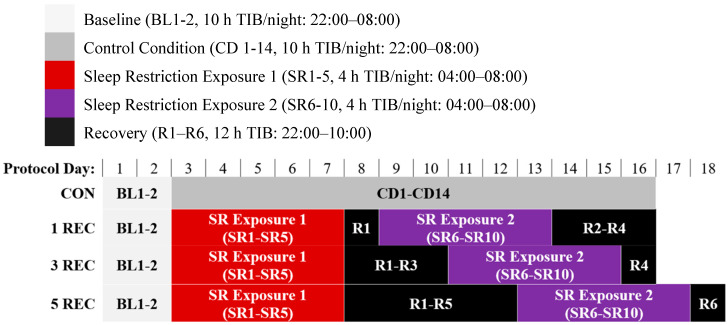
In-laboratory experimental protocol schematic. After two nights of baseline sleep (BL1-2; 22:00–08:00), participants were randomly assigned to one of the following four conditions: 1. CON (Control): 14 nights of 10 h time-in-bed (TIB) per night (22:00–08:00; CD 1-14); 2. 1 REC: five nights of sleep restriction (SR1–SR5, 4 h TIB/night, 04:00–08:00), followed by one night of recovery sleep (R1, 12 h TIB, 22:00–10:00), five nights of sleep restriction (SR6–SR10, 4 h TIB/night, 04:00–08:00), and three nights of recovery sleep (R2–R4; 12 h TIB/night, 22:00–10:00); 3. 3 REC: five nights of sleep restriction (SR1–SR5, 4 h TIB/night, 04:00–08:00), followed by three nights of recovery sleep (R1–R3, 12 h TIB/night, 22:00–10:00), five nights of sleep restriction (SR6–SR10, 4 h TIB/night, 04:00–08:00), and one night of recovery sleep (R4; 12 h TIB, 22:00–10:00); or 4. 5 REC: five nights of sleep restriction (SR1–SR5, 4 h TIB/night, 04:00–08:00), followed by five nights of recovery sleep (R1–R5, 12 h TIB/night, 22:00–10:00), five nights of sleep restriction (SR6–SR10, 4 h TIB/night, 04:00–08:00), and one night of recovery sleep (R6; 12 h TIB, 22:00–10:00).

**Figure 2 nutrients-12-02694-f002:**
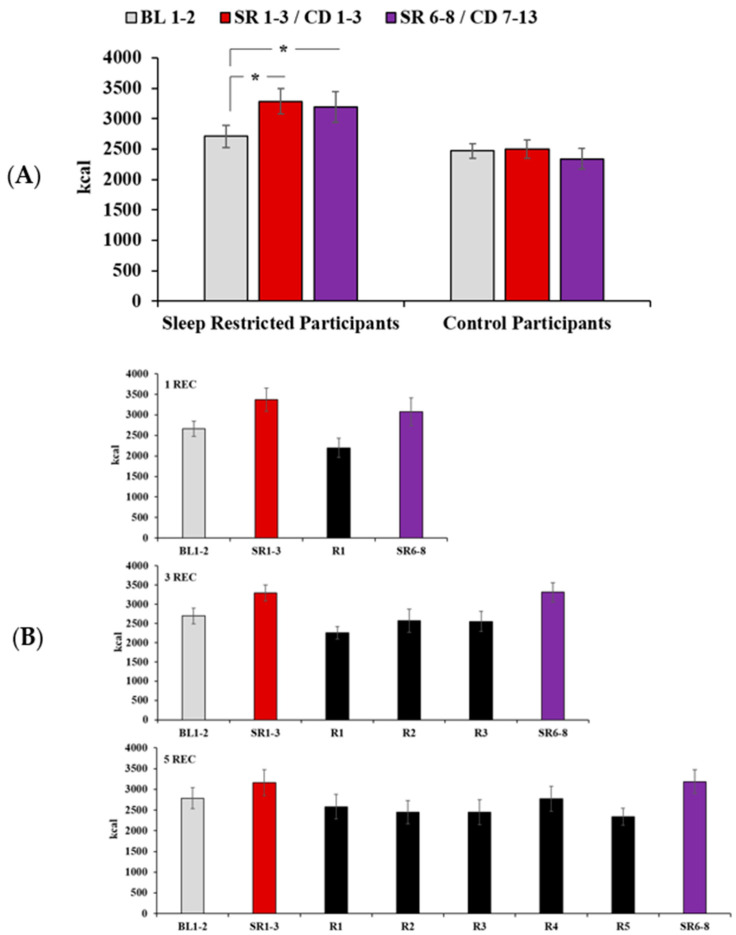
Caloric intake across the protocol in sleep-restricted and control participants. (**A**) Compared to baseline (BL1-2), sleep-restricted participants increased caloric intake during both exposures of sleep restriction (SR1-3: *p* < 0.001, d = 1.3; SR6-8: *p* < 0.001, d = 0.84). Caloric intake did not differ between sleep restriction exposures (*p* = 0.18). Caloric intake did not vary across corresponding days among participants in the control condition (CD1-3, CD7-13). (**B**) There was not a significant Day*Experimental Condition interaction (*p* = 0.27) for caloric intake. The increase in caloric intake observed during both exposures of sleep restriction, relative to baseline intake, was similar across experimental condition groups (1 REC, 3 REC, 5 REC) with varying intervening recovery nights (R1, R1-3, R1-5). Data presented as mean ± SEM, * *p* < 0.05.

**Figure 3 nutrients-12-02694-f003:**
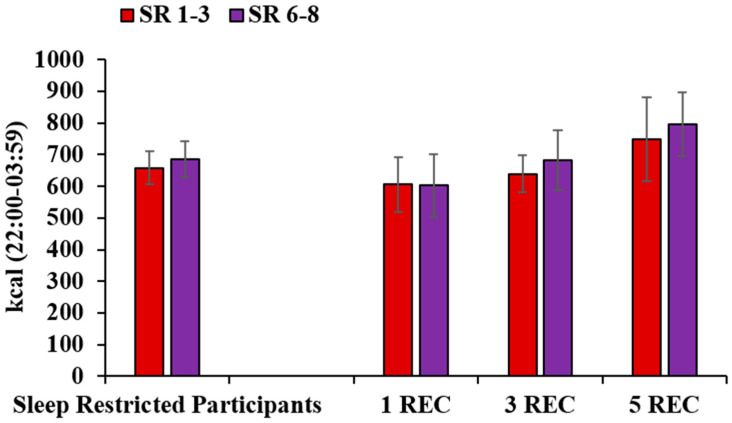
Late-night caloric intake across the protocol in sleep-restricted participants. Late-night caloric intake (mean ± SEM) from 22:00–03:59 did not significantly differ between the first (SR1-3) and second (SR6-8) sleep restriction exposures and there was no significant effect of experimental condition (1 REC, 3 REC, 5 REC; *p*’s > 0.40).

**Table 1 nutrients-12-02694-t001:** Participant characteristics. Mean ± SD or *f*(%).

		Experimental Conditions		
	CON n = 10	1 REC n = 13	3 REC n = 12	5 REC n = 10
Age (years)	33.6 ± 9.8	31.9 ± 7.2	36.2 ± 10.3	34.0 ± 8.9
Women	4 (40.0)	5 (38.5)	5 (41.7)	3 (30.0)
African Americans	6 (60.0)	8 (61.5)	5 (41.7)	6 (60.0)
Body Mass Index (kg/m^2^)	24.3 ± 3.6	25.2 ± 4.3	25.3 ± 3.3	24.0 ± 3.6
Estimated Caloric Need ^a^	2318.8 ± 330.3	2353.4 ± 422.3	2336.3 ± 447.7	2332.5 ± 287.6
Chronotype ^b^	43.00 ± 4.14	42.92 ± 5.53	41.18 ± 5.31	40.50 ± 7.58
Pre-Study Sleep Duration (h) ^c^	7.98 ± 0.53	7.76 ± 0.50	8.29 ± 0.33	7.89 ± 0.38
Pre-Study Sleep Midpoint (time ± min) ^c^	03:37 ± 39.12	03:32 ± 41.29	03:24 ± 32.47	03:40 ± 36.08

CON = control condition (10 h time-in-bed, 22:00–08:00); REC = 1, 3, or 5 nights of recovery sleep (12 h time-in-bed; 22:00–10:00) between two sleep restriction exposures; ^a^ based on the Harris–Benedict equation [41], ^b^ score on Composite Scale of Morningness and Eveningness [32], ^c^ calculated from seven nights of wrist actigraphy.

**Table 2 nutrients-12-02694-t002:** Macronutrient intake across protocol days and by condition (mean ± SD).

	Protein (% kcal)			Carbohydrate (% kcal)			Fat (% kcal)		
	BL1-2	SR1-3/ CD1-3	SR6-8/ CD7-13	BL1-2	SR1-3/ CD1-3	SR6-8/ CD7-13	BL1-2	SR1-3/ CD1-3	SR6-8/ CD7-13
**CON** **(n = 10)**	14.80 ± 3.90	13.70 ± 3.06	14.28 ± 3.43	59.59 ± 5.53	62.29 ± 5.73	60.94 ± 5.58	27.55 ± 4.89	27.00 ± 3.97	27.84 ± 5.30
**All SR Participants** **(n = 35)**	14.30 ± 2.73	12.84 ± 1.98 *****	13.29 ± 1.92 *****	57.95 ± 5.75	58.50 ± 4.38	57.75 ± 5.56	29.38 ± 4.94	30.53 ± 4.20	30.77 ± 5.16
**1 REC** **(n = 13)**	14.62 ± 1.84	13.06 ± 2.06	13.36 ± 1.73	57.28 ± 7.06	56.72 ± 4.37	57.39 ± 6.77	29.85 ± 6.38	32.11 ± 4.73	31.09 ± 6.20
**3 REC** **(n = 12)**	15.73 ± 3.06	13.23 ± 1.86	14.02 ± 2.08	57.47 ± 5.89	58.64 ± 5.54	57.57 ± 5.54	28.49 ± 4.52	30.09 ± 4.08	30.39 ± 5.11
**5 REC** **(n = 10)**	12.18 ± 2.08	12.08 ± 2.00	12.34 ± 1.73	59.39 ± 3.58	60.64 ± 3.80	58.42 ± 4.21	29.84 ± 3.38	29.02 ± 3.20	30.80 ± 4.12

******p* < 0.05, compared with BL1-2.; BL1-2 = baseline sleep days 1 and 2, SR1-3 and SR6-8 = sleep restriction days 1–3 and 6–8, CD1-3 and 7-13 = control condition days 1–3 and 7–13. CON = control condition; 1, 3, and 5 REC = experimental conditions.

**Table 3 nutrients-12-02694-t003:** Fiber intake across protocol days and by condition (mean ± SD).

	Fiber (g)			Fiber (g/1000 kcal)		
	BL1-2	SR1-3/ CD1-3	SR6-8/ CD7-13	BL1-2	SR1-3/ CD1-3	SR6-8/ CD7-13
**CON** **(n = 10)**	20.23 ± 4.57	24.82 ± 4.05 ***^,†^**	21.01 ± 4.46	8.87 ± 3.34	10.63 ± 3.62	9.65 ± 2.75
**All SR Participants** **(n = 35)**	24.01 ± 9.24	27.12 ± 10.34 *****	25.57 ± 10.31	9.26 ± 4.09	8.48 ± 2.84	8.48 ± 4.15
**1 REC** **(n = 13)**	22.66 ± 8.07	26.34 ± 9.53	24.61 ± 11.12	9.20 ± 5.19	8.29 ± 3.47	8.92 ± 5.75
**3 REC** **(n = 12)**	28.08 ± 9.85	29.05 ± 11.02	27.38 ± 10.58	10.71 ± 3.44	8.66 ± 1.50	8.29 ± 3.27
**5 REC** **(n = 10)**	20.88 ± 9.01	25.83 ± 11.28	24.64 ± 9.65	7.61 ± 2.64	8.50 ± 3.39	8.14 ± 2.74

* Compared to BL1-2, *p* < 0.05; † compared to CD 7-13, *p* < 0.05. BL1-2 = baseline sleep days 1 and 2, SR1-3 and SR6-8 = sleep restriction days 1–3 and 6–8, CD1-3 and 7-13 = control condition days 1–3 and 7–13. CON = control condition; 1, 3, and 5 REC = experimental conditions.

**Table 4 nutrients-12-02694-t004:** Sugar intake across protocol days and by condition (mean ± SD).

	Sugar (g)			Sugar (% kcal)		
	BL1-2	SR1-3/ CD1-3	SR6-8/ CD7-13	BL1-2	SR1-3/ CD1-3	SR6-8/ CD7-13
**CON** **(n = 10)**	193.84 ± 70.08	177.27 ± 79.08	164.02 ± 91.61 *****	30.43 ± 5.99	27.64 ± 7.23	26.96 ± 8.00
**All SR Participants** **(n = 35)**	197.26 ± 671.03	231.92 ± 87.16 *****^,†^	211.12 ± 95.68	28.81 ± 5.86	28.09 ± 6.26 **^†^**	26.07 ± 6.46 *****
**1 REC** **(n = 13)**	182.44 ± 68.28	224.86 ± 99.62	200.23 ± 105.11	27.22 ± 5.54	26.56 ± 7.48	25.31 ± 6.38
**3 REC** **(n = 12)**	184.86 ± 65.88	222.04 ± 51.57	193.55 ± 57.75	26.85 ± 5.04	27.02 ± 4.77	23.75 ± 5.34
**5 REC** **(n = 10)**	231.41 ± 75.61	252.97 ± 107.33	246.37 ± 117.54	33.23 ± 5.23	31.37 ± 5.34	29.84 ± 6.66

* Compared to BL1-2, *p* < 0.05; † compared to SR 6-8, *p* < 0.05. BL1-2 = baseline sleep days 1 and 2, SR1-3 and SR6-8 = sleep restriction days 1–3 and 6–8, CD1-3 and 7-13 = control condition days 1–3 and 7–13. CON = control condition; 1, 3, and 5 REC = experimental conditions.

**Table 5 nutrients-12-02694-t005:** Saturated fat intake across protocol days and by condition (mean ± SD).

	Saturated Fat (g)			Saturated Fat (% kcal)		
	BL1-2	SR1-3/ CD1-3	SR6-8/ CD7-13	BL1-2	SR1-3/ CD1-3	SR6-8/ CD7-13
**CON** **(n = 10)**	27.91 ± 10.51	29.41 ± 10.76	25.01 ± 10.15	10.07 ± 2.47	10.29 ± 2.32	9.54 ± 2.17
**All SR Participants** **(n = 35)**	32.22 ± 13.27	40.56 ± 17.12*	38.20 ± 17.47*	10.41 ± 2.86	10.82 ± 2.84	10.52 ± 3.00
**1 REC** **(n = 13)**	32.15 ± 12.49	45.81 ± 19.06	39.90 ± 21.86	10.64 ± 3.46	11.95 ± 3.14	11.22 ± 3.97
**3 REC** **(n = 12)**	30.61 ± 13.95	39.61 ± 15.30	37.75 ± 16.89	9.85 ± 2.80	10.67 ± 2.88	9.98 ± 2.82
**5 REC** **(n = 10)**	34.18 ± 14.54	34.86 ± 16.10	36.55 ± 12.64	10.77 ± 2.17	9.53 ± 1.86	10.26 ± 1.45

* Compared to BL1-2, *p* < 0.05. BL1-2 = baseline sleep days 1 and 2, SR1-3 and SR6-8 = sleep restriction days 1–3 and 6–8, CD1-3 and 7-13 = control condition days 1–3 and 7–13. CON = control condition; 1, 3, and 5 REC = experimental conditions.

**Table 6 nutrients-12-02694-t006:** Macronutrient intake across recovery days by condition (mean ± SD).

	Protocol Day				
	R1	R2	R3	R4	R5
**Protein (% kcal)**					
**1 REC**	13.94 ± 2.54				
**3 REC**	15.11 ± 2.81	15.21 ± 2.74	13.44 ± 3.46		
**5 REC**	13.00 ± 2.82	13.37 ± 2.39	13.29 ± 2.67	12.31 ± 2.50	12.93 ± 2.37
**Carbohydrate (% kcal)**					
**1 REC**	61.08 ± 9.04 *				
**3 REC**	59.76 ± 8.05	56.83 ± 6.70	59.14 ± 8.89		
**5 REC**	57.76 ± 4.05	57.56 ± 5.82	59.74 ± 4.13	56.48 ± 6.44	58.00 ± 5.11
**Fat (% kcal)**					
**1 REC**	26.7 ± 8.41				
**3 REC**	27.03 ± 7.03	29.37 ± 5.58	29.83 ± 7.21		
**5 REC**	30.59 ± 4.32	30.68 ± 4.22	29.20 ± 3.40	32.46 ± 6.23	29.78 ± 5.57

* Compared to BL 1-2, *p* < 0.05 (data for BL 1-2 are in Table 2). R1-5 = Recovery days 1–5. 1, 3, and 5 REC = experimental conditions.

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
