# Peer review of "Caloric and Macronutrient Intake and Meal Timing Responses to Repeated Sleep Restriction Exposures Separated by Varying Intervening Recovery Nights in Healthy Adults"

_nutrients, 2020, doi:10.3390/nu12092694_

Round 1
Reviewer 1 Report
The current manuscript by Spaeth et al., examines the impact of repeated exposures to sleep restriction on eating patterns (amount, content, and timing) and the influence of recovery sleep on these factors. Specifically, the authors analyzed data from 45 participants while undergoing a strict in-laboratory protocol in which they were randomized to one of 4 conditions; control and two bouts of recurrent sleep restriction (4h time in bed) with either one recovery night, three recovery nights, or five recovery nights in between bouts. The authors report that, similar to previous observations, caloric intake consistently increases during sleep restriction, and this increase is unaffected by recovery sleep between sleep restriction bouts. The authors also report that late-night caloric, fiber, sugar, and fat intakes do not differ between sleep restriction bouts. This is a very interesting and timely examination, as the impact of sleep restriction and the timing of food consumption are of great scientific and public interest. The manuscript is well-written, the study appears to be performed with rigor, and the primary conclusions are supported by the data. I do have several comments that should be addressed.
- While I understand there is interest in examining the eating habits during the actual sleep restriction days, it would be very interesting to know if there were any analysis performed that included comparison of the recovery days? In Figure 2, it appears that the first recovery day decreases in caloric intake, but what about distribution (i.e., timing) and composition of those calories during those recovery days? As the authors have previously reported that energy expenditure decreases in the recovery day after sleep restriction (Ref 22), this could be a worthy examination. Moreover, because metabolic rate was measured during the protocol, including metrics of energy balance would be beneficial as the shortened sleep will also increase energy expenditure during the restriction (and the aforementioned decrease during recovery). Thus, the increase of calories may not be the whole story.
- Within the methods section, it is stated that participants could eat either in a common area or privately in their bedrooms. Were multiple participants studied simultaneously, and if so, could social components of the protocol influence eating behaviors (amount, composition, timing, etc)? One could imagine that being around others could not only influence what or how much a person will eat, but also potentially when (PMID: 19808071), thereby warranting some discussion.
- Along the lines of the comment above, do the authors think that placing participants on a strict sleep-wake schedule, not typically synchronized with their habitual timing, may have altered eating habits, particularly on the baseline days? It appears from Table 1 that participants typically went to sleep ~23:30, yet the baseline days initiated the sleep at 22:00. Could this have artificially lowered caloric intake if a participant were a late-evening snacker? Moreover, elongating the wake episode only in the night, and not on both ends of the sleep opportunity (as done in Ref 23 and 25) could artificially increase caloric intake at night and potentially reduce generalizability.
- Should the amounts of fiber, sugar, and saturated fat be analyzed as a change in percent of daily consumption, similar to how the macronutrients were analyzed as percent of daily energy? Otherwise, because intake is increasing, those metrics may also increase due to overall higher intake and not necessarily choosing foods higher in those outcomes.
- Although all the primary conclusions are supported by the data, it may be beyond the scope of the current data/protocol to correlate these types of recovery sleep to weekend recovery sleep due the structured timing of the recovery sleep opportunities.
Minor
- Did the authors control for menstrual phase of the female participants?
- Was there a hypothesis for fiber, sugar, and saturated fats?
Reviewer 2 Report
This is a thorough, complex, and interesting study from an exceptional group. In the current manuscript, authors sought to examine whether repeated exposures to sleep restriction increased caloric intake, whether there was a cumulative increase, and if number of intervening recovery sleep opportunities had an impact. The authors report that sleep restriction increased caloric intake, specifically late-night intake, and that this effect was not cumulative nor did intervening recovery sleep effect intake. The manuscript is well written and concise and the experiment novel. A few suggestions to improve the manuscript:
- With wake set at the same in all experimental conditions (0800), the middle of the sleep period is shifted between the control and the sleep restriction conditions, 0300 vs 0600, respectively. Further, the pre-study sleep midpoint was roughly 3:30am, as indicated in Table 1. Thus, findings in regards to an increase in caloric intake, specifically late eating, may be due, in part, to sleep restriction but also due to a shift in circadian phase. Please discuss.
- It was noted that participants were permitted to watch television. Did this include ads, including ads for food that could be priming and contribute to subsequent intake?
- It is unclear if participants ate at designated times as indicated in lines 151-152 or were they instructed to eat/drink whenever they wanted as indicated on lines 156-157?
- Was there an effect of communal eating vs those that ate alone in their room?
- While the last meal of the day was considered for analysis in regards to the two sleep restriction exposures, were meals across the day compared for the sleep restriction conditions vs the controls? In general, was there an effect of time in the control condition vs the sleep restriction exposures? More eaten later in the day in sleep restriction exposures while the controls ate more earlier in the day? Lastly, did chronotype or habitual intake pattern contribute to observed patterns during baseline and/or the experimental conditions – those that were habitual late eaters ate even more late when exposed to sleep restriction?
- Does the ~ 650 calorie intake observed during late-night intake in the sleep restricted participants account for the entire increase observed over baseline (and the controls)? If the total increased caloric intake observed in Fig 2A, is due to late night eating – this eating opportunity only existed with the 0400 bedtime (line 153) suggesting that another eating opportunity contributed to the increase and not necessarily sleep restriction. Please clarify.
- In Table 3, all participants ate significantly more sugar (g) during SR1-3/CD1-3 than SR6-8/CD7-13? Also, the difference is saturated fat (g) intake was increased for all participants but not for any of the recovery conditions. Both likely due to the differences between participants in intake noted on lines 372-373?
Round 2
Reviewer 1 Report
I thank the authors for their responsiveness.
Author Response
Thank you.